# Change in Estimated Glomerular Filtration Rate After Direct-Acting Antiviral Treatment in Chronic Hepatitis C Patients

**DOI:** 10.3390/diseases13020026

**Published:** 2025-01-21

**Authors:** Gantogtokh Dashjamts, Amin-Erdene Ganzorig, Yumchinsuren Tsedendorj, Dolgion Daramjav, Enkhmend Khayankhyarvaa, Bolor Ulziitsogt, Otgongerel Nergui, Ganchimeg Dondov, Tegshjargal Badamjav, Tulgaa Lonjid, Chung-Feng Huang, Po-Cheng Liang, Batbold Batsaikhan, Chia-Yen Dai

**Affiliations:** 1Department of Internal Medicine, Institute of Medical Sciences, Ministry of Economy and Development, Ulaanbaatar 14210, Mongolia; gantogtokh.ims@mnums.edu.mn (G.D.); aminerdene.ims@mnums.edu.mn (A.-E.G.); yumchinsuren.ims@mnums.edu.mn (Y.T.); dolgion.ims@mnums.edu.mn (D.D.); enkhmend.ims@mnums.edu.mn (E.K.); bolor.ims@mnums.edu.mn (B.U.); otgongerel.ims@mnums.edu.mn (O.N.); ganchimeg.ims@mnums.edu.mn (G.D.); tegshjargal.ims@mnums.edu.mn (T.B.); tulgaa.ims@mnums.edu.mn (T.L.); 2Department of Biological Sciences, School of Life Sciences, Inner Mongolia University, Hohhot 010031, China; 3Department of Internal Medicine, Kaohsiung Medical University Hospital, Kaohsiung Medical University, Kaohsiung 807378, Taiwan; huangcf@kmu.edu.tw (C.-F.H.); pochengliang@gmal.com (P.-C.L.); 4Department of Occupational and Environmental Medicine, Kaohsiung Medical University Hospital, Kaohsiung Medical University, Kaohsiung 807378, Taiwan; 5Department of Health Research, Graduate School, Mongolian National University of Medical Sciences, Ulaanbaatar 14210, Mongolia; 6Ph.D. Program in Environmental and Occupational Medicine, Drug Development and Value Creation Research Center, Kaohsiung Medical University, Kaohsiung 807378, Taiwan; 7College of Professional Studies, National Pingtung University of Science and Technology, Pingtung 91201, Taiwan

**Keywords:** antiviral therapy, hepatitis C, renal function, chronic kidney disease

## Abstract

Background: Hepatitis C virus (HCV) infection accelerates the progression of chronic kidney disease (CKD), increasing the risk of kidney failure and end-stage renal disease. Direct-acting antiviral (DAA) therapies for HCV infection inhibit viral replication by 95–97%, leading to a sustained virologic response. Our objective was to assess renal function in patients with chronic HCV infection in Taiwan after receiving DAA therapy. Goal: Our study included 4823 patients with HCV infection who were undergoing DAA therapy. Renal function was evaluated by calculating the glomerular filtration rate (eGFR). eGFR assessed at the initiation of the treatment, during treatment, and at 3 months, 6 months, 1 year, and 3 years after completion of treatment. The baseline demographic and laboratory parameters of the study participants were evaluated, and the results were analyzed using statistical methods. Results: The average age of the study participants was 61.35 ± 12.50 years, and 54.5% of were male. The mean of eGFR in baseline and after treatment showed a decrease. Liver fibrosis scores (FIB4, APRI, Fibroscan) and liver function tests were significantly improved after DAA treatment (*p* = 0.001). However, white blood count (5.41 ± 1.7 vs. 5.73 ± 1.9), platelet count (168.04 ± 74.0 vs. 182.11 ± 69.4), and creatinine levels (1.05 ± 1.3 vs. 1.12 ± 1.3) increased after treatment (*p* = 0.001). The number of patients with an eGFR of 60 mL/min/1.73 m^2^ decreased both during and after treatment (*p* < 0.001). Among patients with CKD, eGFR improved after DAA treatment (*n* = 690, 35.93 ± 19.7 vs. 38.71 ± 23.8; 95% CI −3.56–1.98; *p* = 0.001). Logistic regression analysis revealed that renal function improved in patients with CKD who had an eGFR of less than 60 mL/min/1.73 m^2^ before DAA treatment (OR 1.62, 95% CI 1.37–1.91, *p* = 0.001). Conclusions: In individuals with CKD and a baseline eGFR < 60 mL/min per 1.73 m^2^, eGFR level was increased during DAA treatment. This suggests that initiating DAA therapy in HCV-infected patients, even those without clinical manifestations, could be a crucial strategy to prevent further decline in renal function.

## 1. Introduction

According to the World Health Organization, around 50 million people are infected with hepatitis C virus (HCV) globally, and in 2022, over 242,000 people died from cirrhosis and hepatocellular carcinoma [1]. Although HCV infections have been steadily declining globally since 2015 [2,3,4], geographically distinct countries such as Taiwan and Mongolia continue to experience significant rates of HCV infection.

Studies have established that HCV infection leads to chronic changes in other organ systems, in addition to causing liver inflammation [5]. The most common HCV-related nephropathy is membranoproliferative glomerulonephritis (MPGN), which is typically associated with cryoglobulinemia. Although studies have shown that proteinuria occurs in HCV infection, the results regarding its effect on the glomerular filtration rate (GFR) in the kidneys may vary depending on the HCV genotype, the climate of the country, and other contributing factors.

A study in Taiwan found that 16.5% of HCV-infected patients had chronic kidney disease (CKD) [6]. The prevalence of HCV-related complications is high among patients with end-stage renal disease (ESRD) and severe renal impairment [7,8,9].

HCV infection in patients on hemodialysis or with ESRD is associated with a higher risk of death, increased hospitalization rates, and a lower quality of life [10]. Consequently, direct-acting antiviral (DAA) treatment is urgently needed in these populations. Given the direct and indirect effects of HCV on other organ systems, it is hypothesized that HCV may influence renal function and metabolism, especially after DAA therapy. Over the past 5 years, HCV has become a curable illness due to the advent of DAA therapies. DAAs target viral proteins essential for HCV replication and, when used in combination, achieve cure rates exceeding 97% [11,12,13]. Despite these advances, the effects of DAA treatment on both short- and long-term kidney function remain poorly understood. Therefore, it is believed that successful HCV eradication could improve HCV-related renal dysfunction [14]. However, a recent large study reported that successful HCV eradication with DAAs may actually result in worsening renal function [8]. It remains unclear whether the association between HCV and rapid CKD progression can be mitigated by DAA therapy [15]. This study aimed to assess renal function before and after DAA therapy, as well as to investigate the baseline characteristics, laboratory parameters, and comorbidities of the study participants.

## 2. Materials and Methods

This study included patients who had not received anti-HCV treatment at Kaohsiung Medical University Hospital between 2004 and 2021. Patients were treated with interferon-free direct-acting antivirals (DAAs). The choice of antiviral regimen was based on the HCV treatment guidelines of the Asian Pacific Association for the Study of the Liver (APASL) and the reimbursement criteria established by the National Health Insurance Administration, Taiwan. The primary objective was to achieve an SVR at 12 weeks, defined as maintaining an undetectable level of HCV ribonucleic acid (RNA) (<12 IU/mL or <25 IU/mL, depending on the laboratory test used) during the 12-week post-treatment follow-up (SVR12) period. CKD was defined as the presence of proteinuria in the urine for more than 3 months or an eGFR of less than 60 mL/min/1.73 m^2^.

Data were collected retrospectively from the medical records, including age, sex, body mass index (BMI), presence of type 2 diabetes mellitus, hypertension, hyperlipidemia, alcohol drinking, smoking history, serum biochemistry, and AFP. The degree of liver fibrosis was assessed using noninvasive methods such as the FIB4 index and APRI and invasive methods like Fibroscan with acoustic radiation pulse elastography > 1.81 m/s, or histologically confirmed F3/4.

Exclusion criteria included co-infection with hepatitis B or human immunodeficiency virus, history of the prior presence of HCC, prior treatment for HCV infection, liver transplantation, other malignancy, alcoholism, idiopathic hepatic fibrosis, and hepatic fibrosis caused by other secondary conditions. Serum cryoprecipitation was conducted using a previously published method [15,16,17]. Renal function was evaluated by calculating the eGFR. eGFR is a standard method for assessing CKD based on diagnostic criteria, and it defines 5 stages of kidney disease, independent of the amount of protein in the urine.

### Statistical Analyses

We calculated mean values and standard deviations for continuous variables. To assess the differences between variables, independent *t*-tests and Chi-square tests were used. To evaluate the relationship between associated factors and post DAA treatment renal function, we conducted both univariate analysis and multivariate logistic regression analysis. The analysis of the research results was performed using SPSS software (version 26.0, SPSS Inc., Chicago, IL, USA). Differences between groups were considered statistically significant if *p* < 0.05.

## 3. Results

Baseline characteristics of all participants before starting DAA therapy were compared with those at 3 months after completing DAA treatment. Before starting DAA patients’ mean age was 61.36 ± 12.3, and three months after DAA, their mean age was 61.82 ± 12.3. There was no statistical difference in body mass index before and after starting DAA (24.8 ± 3.3 vs. 25.2 ± 3.8, *p* < 0.259). The number of cases of diabetes, arterial hypertension, hyperlipidemia, drinking alcohol, and smoking history among patients decreased after the start of DAA (*p* = 0.001). The results showed that after DAA treatment, there were increases in white blood cell count (6.01 ± 2.0/SD, *p* = 0.001) and platelet count (192.12 ± 71.8, *p* = 0.001). Additionally, AST and ALT levels decreased (28.98 ± 16.7; 23.84 ± 20.0, *p* = 0.001), as did direct bilirubin levels (0.16 ± 0.1, *p* = 0.001). Furthermore, following DAA treatment, hepatic inflammatory activity decreased, platelet count increased, cytolysis improved, and direct bilirubin decreased, reflecting improved hepatic function. After DAA patients’ levels of alpha-fetoprotein increased, however, there was no statistically significant improvement (16.58 ± 279.2 vs. 56.04 ± 2649.7, *p* < 0.279) (Table 1).

Baseline total cholesterol and triglyceride levels (187.79 ± 40.4; 110.91 ± 72.3, *p* = 0.001) were significantly higher at 3 months after DAA treatment compared to baseline levels. There was no significant difference in mean eGFR before the initiation of DAA treatment and mean eGFR 3 months after completing DAA treatment for all participants. However, compared to baseline hepatic fibrosis grade (as assessed by Fibroscan median, FIB4, and APRI), the mean hepatic fibrosis grade decreased 3 months after the DAA treatment (3.23 ± 3.0 vs. 2.49 ± 2.1, *p* = 0.001). Additionally, the number of patients with cryoglobulinemia decreased significantly from baseline to 3 months after DAA treatment (821 (15.99%) vs. 70 (1.4%), *p* = 0.001). Moreover, when all participants were classified into five stages of CKD, the number of patients with CKD decreased from baseline to 3 months after DAA treatment (Table 1).

Laboratory parameters differed significantly (*p* = 0.001) in participants with eGFR < 60 mL/min/1.73 m^2^ between baseline (before starting DAA) and at 3 months after completing DAA treatment. Participants with an eGFR < 60 mL/min/1.73 m^2^ showed an increase in mean eGFR at 3 months after DAA treatment, compared to baseline (eGFR < 60 mL/min per 1.73 m^2^ 35.93 ± 19.7 vs. 38.71 ± 23.8; *p* = 0.01) (Table 2). After DAA completion, the median hepatic fibrosis score (as assessed by Fibroscan, FIB4, and APRI) significantly decreased (*p* = 0.001). Moreover, the number of patients with cryoglobulinemia and CKD also decreased significantly (*p* = 0.001) (Table 2).

According to univariate logistic regression analysis, diabetes mellitus and male gender were associated with a decrease in eGFR after DAA treatment. However, among individuals with an eGFR < 60 mL/min/1.73 m^2^ prior to DAA treatment, the likelihood of an increase in eGFR after DAA treatment was 1.6 times higher. After DAA treatment, elevated eGFR was not associated with age, the degree of liver fibrosis (F3–F4), or arterial hypertension (Table 3).

After DAA treatment, patients with a baseline eGFR of more than 60 mL/min/1.73 m^2^ experienced a statistically significant incrise compared to those with a baseline eGFR of less than 60 mL/min/1.73 m^2^ (Table 4).

As shown in Figure 1, the number of patients with diabetes mellitus, arterial hypertension, CKD, cryoglobulinemia, and proteinuria before the start of DAA treatment is compared with the number of patients diagnosed at the end of DAA treatment and at follow-up 1, 2, and 3 years after treatment. The number of patients with comorbidities decreases as time after completion of DAA treatment increases (*p* < 0.001, Figure 1).

Furthermore, 159 patients with diabetes mellitus, 113 patients with cryoglobulinemia, 84 patients with arterial hypertension, 158 patients with proteinuria, and 146 patients with other organ tumors did not have these comorbidities after DAA treatment. However, 105 patients with diabetes mellitus, 11 patients with cryoglobulinemia, 16 patients with arterial hypertension, 39 patients with proteinuria, and 10 patients with other organ tumors were registered with comorbidities after DAA treatment, despite not having them before (Table 5).

During the follow-up period after the completion of DAA treatment, hepatocellular carcinoma was diagnosed in 27 patients during first year, 17 patients in the second year, and 10 patients in the third year. Tumors of other organs were diagnosed in 32 patients in the first year, in 9 patients in the second year, and in 2 patients in the third year. Hyperlipidemia was observed in 32 patients in the first year, in 9 patients in the second year, and in 2 patients in the third year.

Three patients required hemodialysis during the first year after the end of DAA treatment but did not need hemodialysis at the second-year follow-up. Additionally, two new cases of cardiac infarction were diagnosed after DAA therapy. There were a number of patients in whom the status of comorbidities increased during the follow-up period, both before and after DAA treatment.

When participants were compared before and after DAA treatment according to comorbidities, eGFR in patients with diabetes mellitus, hyperlipidemia, and arterial hypertension increased after DAA treatment compared to pre-DAA treatment (*p* < 0.01). However, in patients without diabetes, hyperlipidemia, and arterial hypertension, the rate of eGFR decreased after DAA treatment (*p* < 0.01), (Table 6). Additionally, eGFR decreased after DAA treatment in both groups of patients with or without cryoglobulinemia and proteinuria, as well as in patients with a hepatic fibrosis score (FIB4) greater than 3.25 (*p* < 0.01). Notably, in patients with cryoglobulinemia and proteinuria, eGFR significantly decreased after DAA treatment.

Participants with CKD were divided into two groups, diabetic and nondiabetic, before and after DAA treatment, and their eGFRs were compared. It was observed that the eGFR in nondiabetic participants was higher than in diabetic participants before DAA treatment, but it decreased after DAA treatment (Figure 2).

## 4. Discussion

This research is among the first to assess kidney function in patients receiving treatment with DAAs. Upon evaluating the impact of DAAs on the rate of eGFR decline, we observed that patients with an eGFR < 60 mL/min/1.73 m^2^ experienced a notable stabilization in eGFR decline following DAA treatment. In addition, we found that DAA therapy was associated with an improvement in eGFR, particularly in patients with CKD. Earlier histopathologic studies have indicated that glomerular abnormalities are observed in 55–85% of unselected patients with HCV infection at the time of liver transplantation or death [18,19]. A study investigating the effects of sofosbuvir-based DAAs found that kidney function deteriorated more significantly in patients with stage 3a CKD [13]. However, this study defined “worsening renal function” using text-based diagnosis codes from adverse event reports, whereas our study utilized raw creatinine values and staged CKD events according to established diagnostic criteria.

Previous studies of patients treated with interferon-based regimens suggested that HCV therapy could decrease the case of CKD and slow the progression to ESRD. For instance, Satapathy et al. [19] found that the prevalence of CKD was higher in the HCV-infected group than in the control group. This suggests that HCV infection accelerates the progression of CKD and leads to ESRD. This mechanism may be related to both direct and indirect effects of HCV exposure on the kidney. Additionally, two national cohort studies from Taiwan demonstrated that interferon-based treatment led to a 58% reduction in the incidence of CKD and an 84% decrease in the risk of ESRD [20,21]. However, interferon-based regimens, which formed the backbone of HCV treatment before the advent of DAAs, were poorly tolerated, had low sustained virologic response (SVR) rates, and posed challenges for CKD patients due to the renal elimination of ribavirin.

As a result, only a small number of CKD patients received interferon-based therapies. The advent of DAA regimens provided the first opportunity to comprehensively assess the impact of HCV eradication on kidney function, as these treatments are now safe and effective for a large number of patients. Small studies indicate that DAAs may improve kidney function in HCV-infected patients. For example, patients undergoing HCV treatment and liver transplantation share common features of CKD [22]. In patients with stage I-II CKD, achieving SVR after DAA treatment positively affects renal function. However, in our study, an increase in GFR was observed after DAA treatment in patients with an eGFR < 60 mL/min/1.73 m^2^. In another study, renal function remained stable during DAA treatment. Sofosbuvir-based DAA therapy was effective in patients with CKD up to stage 3a (eGFR 45–59 mL/min/1.73 m^2^), as shown by a multivariable model [23].

Our study found that in individuals with eGFR < 60 mL/min/1.73 m^2^ prior to DAA treatment, the likelihood of an increase in eGFR after DAA treatment was 2.78 times higher. DAA treatment has been shown to cause mild to moderate side effects, particularly in patients with ESRD [24]; additionally, none of the patients discontinued DAAs during this study. However, in real-world practice, very few CKD patients received DAAs for HCV treatment. A few patients with HCV infection and CKD have received DAA therapy. The introduction of DAA therapy for HCV infection has provided an interferon-free treatment option with fewer side effects than interferon therapy and a higher rate of SVR [25]. DAA therapy may reduce major complications in patients with CKD. DAA therapy aims to reduce cardiovascular complications and mortality in patients with HCV infection and kidney failure [26]. Patients with HCV infection who received DAA therapy had a reduced risk of developing kidney failure, stroke, and cardiovascular disease compared to those who did not receive treatment. In addition, treating HCV infection reduced vascular complications in patients with diabetes [27,28]. This could be explained by the mechanisms of atherosclerosis caused by HCV infection and insulin resistance.

In our study, fewer patients developed myocardial infarction and arterial hypertension following DAA treatment. DAA treatment slows the progression of kidney function loss in patients with CKD, reduces liver inflammation, and improves patients’ quality of life [29]. Previous studies have shown that the use of DAAs in patients with HCV-associated glomerulonephritis resulted in increased SVR rates, improved serum creatinine levels, and reduced proteinuria [23]. It is considered necessary to monitor further, as the development of an SVR against HCV with DAA therapy improves overall body metabolism (including glucose metabolism) and kidney function and increases lipid metabolism [30]. Notably, 159 participants in our study had diabetes prior to antiviral treatment but did not have diabetes after treatment.

Gantsetseg et al.’s [31] study showed that HCV-infected patients were at a higher risk of decreased renal eGFR compared to healthy controls. In addition, obesity, diabetes, and hypertension are associated with a greater risk of renal dysfunction. In our study, however, patients with diabetes, hyperlipidemia, and hypertension showed an increased eGFR after DAA treatment. Cryoglobulinemia, proteinuria, and advanced fibrosis were associated with a decrease in eGFR after DAA treatment. Lipid levels increased significantly after DAA treatment and were not associated with changes in lipid levels after treatment for advanced fibrosis. Changes in lipid profiles following HCV clearance have been reported in several studies [32,33,34,35], which also noted significant increases in LDL levels post-treatment.

In our study, renal function improved during follow-up after HCV eradication with DAA therapy. Specifically, patients with eGFR 60 < ml/min/1.73 m^2^ showed an improvement in serum creatinine and a 10% increase in eGFR compared to the baseline. Multivariate analysis revealed that DAAs were an independent predictor of eGFR improvement, whereas age had no significant effect (Table 3).

HCV infection can lead to kidney disease and even chronic kidney failure. According to a study by Mbaeyi et al., the frequency of kidney dialysis in HCV-infected patients in the United States is 7–8 times higher than the average for the general population [36]. The mechanism of renal injury caused by HCV infection is not fully understood, but researchers have proposed several hypotheses. Most reports suggest that the pathophysiology depends on renal infection or host immune mechanisms. The pathogenesis is associated with viral infection through both immune and non-immune mechanisms [37]. DAA therapy is associated with an SVR in patients with CKD or cirrhosis [38]. HCV eradication has been shown to be important for preventing extrahepatic complications of HCV [39].

Our study supports these findings, demonstrating a significant improvement in renal function after HCV clearance, particularly in patients with an eGFR < 60 mL/min/1.73 m^2^. However, in patients with eGFR > 60 mL/min/1.73 m^2^, no significant changes in renal function were observed after treatment.

## 5. Conclusions

Our study demonstrates that DAAs can slow the rate at which patients with compromised renal function (eGFR < 60 mL/min/1.73 m^2^) experience a decline in glomerular filtration rate. This is the first analysis suggesting that DAAs may slow the rapid decline in eGFR associated with chronic HCV infection. The results emphasize the importance of HCV eradication not only in preventing the progression of liver disease but also in improving or preventing renal and extrahepatic manifestations that contribute significantly to morbidity and mortality in these patients.

## Figures and Tables

**Figure 1 diseases-13-00026-f001:**
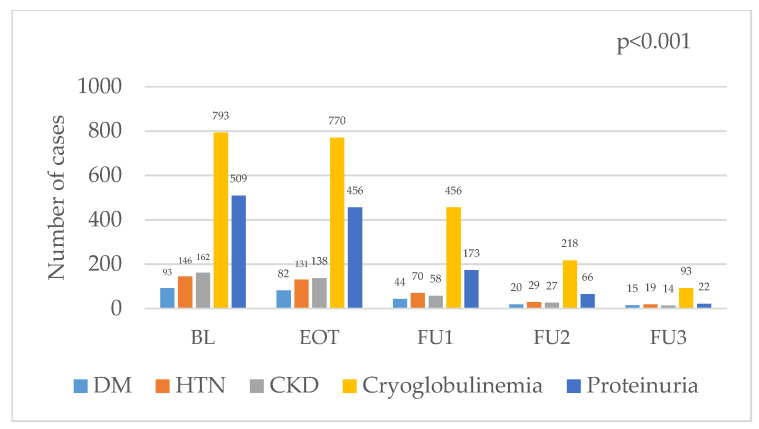
The number of patients with comorbidities before, at the end of, and after the completion of DAA treatment.

**Figure 2 diseases-13-00026-f002:**
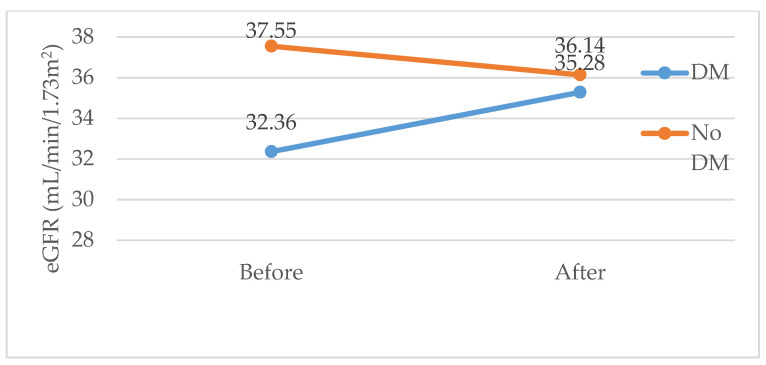
The evaluation and comparison of glomerular filtration rate (GFR) in diabetic and nondiabetic participants were conducted before and after DAA treatment.

**Table 1 diseases-13-00026-t001:** Baseline characteristics compared with the characteristics after 3 months of completion DAA treatment.

Parameters	Baseline Characteristics	After 3 Months Characteristics	*p* Value
Age (mean ± SD)	61.36 ± 12.3	61.82 ± 12.3	0.001
Sex, *n* (%)			
Male	2395 (46.4)	1990 (45.3)	0.162
Female	2766 (53.6)	2404 (54.7)	
BMI	24.8 ± 3.3	25.2 ± 3.8	0.259
Presence of type 2 diabetes mellitus, *n* (%)			
Yes	97 (1.9)	14 (0.3)	0.001
No	3872 (75.0)	2011 (39.0)	
Hypertension, *n* (%)			
Yes	166 (3.2)	21 (0.4)	0.001
No	2935 (56.9)	2003 (38.8)	
Hyperlipidemia, *n* (%)			
Yes	258 (5.0)	32 (0.6)	0.001
No	4221 (81.8)	1993 (38.6)	
Alcohol drinking, *n* (%)	655 (12.7)	458 (8.2)	0.001
Smoking history, *n* (%)	1013 (19.6)	885 (16.5)	0.001
**Serum biochemistry**			
WBC (mean ± SD)	5.81 ± 1.9	6.01 ± 2.0	0.001
HGB (mean ± SD)	13.49 ± 1.9	13.51 ± 1.9	0.336
PLT (mean ± SD)	186.30 ± 72.0	192.12 ± 71.8	0.001
AST (mean ± SD)	61.62 ± 47.1	28.98 ± 16.7	0.001
ALT (mean ± SD)	70.72 ± 63.3	23.84 ± 20.0	0.001
BIl (D) (mean ± SD)	0.20 ± 0.2	0.16 ± 0.1	0.001
AFP (mean ± SD)	16.58 ± 279.2	56.04 ± 2649.7	0.279
eGFR, mL/min per 1.73 m^2^	83.92 ± 30.1	83.62 ± 30.6	0.265
CHOL (T) (mean ± SD)	172.77 ± 38.0	187.79 ± 40.4	0.001
TG (mean ± SD)	101.98 ± 56.7	110.91 ± 72.3	0.001
Fibroscan median	12.42 ± 10.9	9.24 ± 8.1	0.001
FIB4	3.23 ± 3.0	2.49 ± 2.1	0.001
APRI	1.11 ± 1.3	0.48 ± 0.5	0.001
Cryoglobulinemia, *n* (%)	821 (15.9)	70 (1.4)	0.001
CKD, *n* (%)	4823	3996	0.001
I	2013 (39.0)	1616 (31.3)	
II	2015 (39.1)	1683 (32.6)	
III	517 (10.0)	455 (8.8)	
IV	62 (1.2)	22 (0.8)	
IV	216 (4.2)	200 (3.9)	

WBC: white blood count; HGB: hemoglobin; PLT: platelet; AST: aspartate aminotransferase; ALT: alanine aminotransferase; BIl: bilirubin; AFP: alpha fetoprotein; eGFR: estimated glomerular filtration rate; CHOL: cholesterol; TG: triglycerides; FIB4: index for liver fibrosis; APRI: AST-to-platelet ratio index; CKD: chronic kidney disease.

**Table 2 diseases-13-00026-t002:** Baseline characteristics versus 3 months after completion of DAA treatment in participants with eGFR < 60 mL/min/1.73 m^2^.

Parameters	Baseline Characteristics	After 3 Months Characteristics	*p* Value
Age (mean ± SD)	69.16 ± 10.4	69.63 ± 10.5	0.001
WBC (mean ± SD)	6.04 ± 2.0	6.27 ± 2.0	0.001
HGB (mean ± SD)	12.08 ± 2.0	12.17 ± 2.0	0.077
PLT (mean ± SD)	178.42 ± 71.3	182.56 ± 70.8	0.034
AST (mean ± SD)	54.48 ± 50.2	26.98 ± 13.2	0.001
ALT (mean ± SD)	53.46 ± 52.3	20.65 ± 23.8	0.001
BIl (D) (mean ± SD)	0.18 ± 0.1	0.15 ± 0.1	0.001
AFP (mean ± SD)	11.13 ± 71.5	24.93 ± 234.6	0.082
GFR mL/min per 1.73 m^2^	35.93 ± 19.7	38.71 ± 23.8	0.001
CHOL (T) (mean ± SD)	164.48 ± 37.5	181.41 ± 43.6	0.001
TG (mean ± SD)	115.13 ± 59.3	124.13 ± 70.6	0.001
Fibroscan median	13.56 ± 11.2	10.44 ± 10.4	0.001
FIB4	3.65 ± 3.4	2.92 ± 2.3	0.001
APRI	1.00 ± 1.4	0.46 ± 0.4	0.001
Cryoglobulinemia, *n* (%)	109 (13.7)	43 (5.4)	0.001
CKD, *n* (%)	*n* = 793	*n* = 689	0.001
I			
II			
III	515 (64.9)	331 (41.7)	
IV	62 (7.8)	41 (5.2)	
IV	216 (27.2)	196 (24.7)	

WBC: white blood count; HGB: hemoglobin; PLT: platelet; AST: aspartate aminotransferase; ALT: alanine aminotransferase; BIl: bilirubin; eGFR: estimated glomerular filtration rate; CHOL: cholesterol; TG: triglycerides; FIB4: index for liver fibrosis; APRI: AST-to-platelet ratio index; CKD: chronic kidney disease.

**Table 3 diseases-13-00026-t003:** Univariate and multivariable logistic regression model predicting eGFR improvement after DAA therapy.

	Univariate Model			Multivariate Model		
Baseline Predictors of eGFR Improvement After DAAs	OR	95% CI	*p* Value	OR	95% CI	*p* Value
Age, per 50 yr	0.90	0.77–1.05	0.20			
Sex (female vs. male)	0.87	0.77–0.99	0.03	0.88	0.78–0.99	0.03
Nondiabetic vs. diabetic	0.83	0.71–0.96	0.01	0.75	0.64–0.87	0.01
Cirrhosis (vs. noncirrhotic)	0.93	0.82–1.06	0.29			
Hypertension (vs. non-hypertensive)	0.95	0.84–1.08	0.48			
Baseline eGFR < 60 mL/min per 1.73 m^2^	1.62	1.37–1.91	0.01	1.73	1.46–2.05	0.01

CI: confidence interval; DAA: direct-acting antiviral; eGFR: estimated glomerular filtration rate. eGFR improvement was defined by >10% increase in eGFR from baseline to post-treatment average in the 12 months after completing DAA therapy. The multivariable model includes demographics and predictors with *p* < 0.1 in the univariate model. Only nondiabetic status and baseline eGFR < 60 mL/min per 1.73 m^2^ predicted improvement in eGFR after DAAs in multivariable models.

**Table 4 diseases-13-00026-t004:** Factors associated with eGFR increase of more than 10 mL/min/1.73 m^2^.

Characteristics	OR	95% CI	*p* Value
Age, per 50 yr	1.12	0.93–1.35	0.21
Sex (female vs. male)	0.98	0.84–1.13	0.81
Nondiabetic patients vs. diabetic	0.89	0.75–1.06	0.20
Cirrhosis (vs. noncirrhotic) FIB4_3.25	0.85	0.73–0.99	0.04
Non-HTN vs. HTN	1.12	0.96–1.30	0.12
Baseline eGFR < 60 mL/min per 1.73 m^2^	5.52	3.96–7.70	0.01

OR: odds ratio; CI: confidence interval; DAA: direct-acting antiviral; HTN: hypertension; eGFR: estimated glomerular filtration rate. eGFR measurements taken from baseline to 3 months after DAA therapy were included in the analysis. The generalized estimating equation was used to derive the slope estimates after adjusting for age, baseline eGFR, and diabetes.

**Table 5 diseases-13-00026-t005:** Number of patients with and without comorbidities before and after DAA therapy.

Baseline	Post Treatment	Participants	*p* Value
Nondiabetic vs. diabetic patients	Nondiabetic vs. diabetic patients		*p* < 0.001
−	−	2691	
+	−	159	
−	+	105	
+	+	382	
Non-cryoglobulinemia vs. cryoglobulinemia patients	Non-cryoglobulinemia vs. cryoglobulinemia patients		*p* < 0.001
−	−	252	
+	−	113	
−	+	11	
+	+	40	
Non-hypertension vs. hypertension patients	Non-hypertension vs. hypertension patients		*p* < 0.001
−	−	1137	
+	−	84	
−	+	16	
+	+	0	
Non-proteinuria vs. proteinuria patients	Non-proteinuria vs. proteinuria patients		*p* < 0.001
−	−	144	
+	−	158	
−	+	39	
+	+	176	
Non-other cancer vs. other cancer patients	Non-other cancer vs. other cancer patients		*p* < 0.001
−	−	1868	
+	−	146	
−	+	10	
+	+	1	

(−): negative; (+): positive

**Table 6 diseases-13-00026-t006:** Comparison of eGFR in participants with and without comorbidity before and after DAA treatment.

Parameters	Total	Baseline eGFR Level, Mean ± SD	Total	After Treatment eGFR Level, Mean ± SD	*p* Value
DM, *n*					0.001
Yes	1044	73.95 ± 34.9	117	81.42 ± 29.8	
No	3779	86.70 ± 28.7	1901	82.86 ± 29.07	
HDL, *n*					0.001
Yes	739	76.82 ± 32.4	136	81.54 ± 30.5	
No	4084	85.22 ± 30.1	1882	82.86 ± 290	
HTN, *n*					0.001
Yes	1964	73.33 ± 31.1	126	83.14 ± 29.2	
No	2859	91.23 ± 27.8	1892	82.74 ± 29.1	
Cryoglobulinemia, *n*					0.001
Yes	3183	85.08 ± 31.1	21	47.04 ± 36.2	
No	1640	81.72 ± 29.5	244	53.14 ± 31.5	
Proteinuria, *n*					0.001
Yes	2197	78.25 ± 35.4	240	39.72 ± 31.3	
No	2626	88.70 ± 24.8	164	65.50 ± 25.9	
FIB4					0.001
<3.25	3260	85.24 ± 30.8	1533	84.23 ± 28.6	
>3.25	1548	81.15 ± 29.9	448	78.53 ± 30.4	

SD: standard deviation; DM: diabetes mellitus; HDL: hyperlipidemia; HTN: arterial hypertension; FIB4: fibrosis index.

## Data Availability

Data are contained within the article.

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
