# Peer review of "Change in Estimated Glomerular Filtration Rate After Direct-Acting Antiviral Treatment in Chronic Hepatitis C Patients"

_diseases, 2025, doi:10.3390/diseases13020026_

Round 1

Reviewer 1 Report

Comments and Suggestions for Authors

The article “An estimated glomerular filtration rate elevation after direct-acting antiviral treatment in chronic hepatitis C patients” explores an interesting and relevant topic. However, the following aspects require improvement:

   1. The title does not accurately reflect the study results and needs to be revised for clarity and alignment with the findings.
   2. The introduction requires further refinement to enhance structure and focus.
   3. The similarity index is excessively high and must be reduced to meet acceptable standards.
   4. Clarification is needed regarding the treatment regimens used in the study cohort (DAAs or interferon-based therapy), as this appears inconsistent with the stated study objectives.

Author Response

Reviewer #1: General comments

The article “An estimated glomerular filtration rate elevation after direct-acting antiviral treatment in chronic hepatitis C patients” explores an interesting and relevant topic. However, the following aspects require improvement:

Specific comments

  1. The title does not accurately reflect the study results and needs to be revised for clarity and alignment with the findings?

Thank you very much for this comment. The title of the study has been changed to "The change of an estimated glomerular filtration rate after direct acting antiviral treatment in chronic hepatitis C patients".

  1. The introduction requires further refinement to enhance structure and focus?

Thank you very much for this reminder and we have added this information in introduction section and all changes appears in different color.

  1. The similarity index is excessively high and must be reduced to meet acceptable standards?

Thank you for this comment. As for the similarity we have revised our manuscript and have made several changes, almost 30% of the manuscript has been changed. Please find the revised changes in different color.

  1. Clarification is needed regarding the treatment regimens used in the study cohort (DAAs or interferon-based therapy), as this appears inconsistent with the stated study objectives?

Thank you very much for this comment. The choice of antiviral regimen was based on the HCV treatment guidelines of the Asian Pacific Association for the Study of the Liver (APASL) and the reimbursement criteria set by the National Health Insurance Administration, Taiwan. Also we have added this information in Materials and Methods section.

Reviewer 2 Report

Comments and Suggestions for Authors

The manuscript is devoted to researching the renal function in patients with chronic HCV infection. Renal diseases are the most common extrahepatic complications associated with HCV infection and affect 10-60% of patients. The study is very important, it allows us to assess the danger of chronic kidney disease in patients with hepatitis C in Taiwan.

Analysis was made on a basis of characteristics after 3 months of therapy, age, the degree of liver fibrosis, hyperlipidemia and arterial hypertension were taken into account.

Treatment with DAAs provide several benefits for CKD patients with HCV infection: reduction in cardiovascular morbidity and mortality, moreover progression of renal dysfunction can be achieved, reducing liver disease progression, and improving patients’ well-being.

It was demonstrated that DAA therapy can improve renal function in HCV patients, particularly those with baseline eGFR below 60 ml/min per1.73m². This finding suggests that early intervention with DAAs may help prevent further deterioration of kidney function, emphasizing the need for timely diagnosis and treatment of HCV infection in patients with chronic kidney disease.

Data is discussed well and is of interest to a wide range of readers

I would recommend minor revisions.

1)      Direct-acting antivirals used for the treatment are not mentioned. Did the authors mean to analyze the effect regardless of the specific antiviral drug?

References 46-48 seem strange

Author Response

Reviewer #2: General comments

The manuscript is devoted to researching the renal function in patients with chronic HCV infection. Renal diseases are the most common extrahepatic complications associated with HCV infection and affect 10-60% of patients. The study is very important, it allows us to assess the danger of chronic kidney disease in patients with hepatitis C in Taiwan.

Analysis was made on a basis of characteristics after 3 months of therapy, age, the degree of liver fibrosis, hyperlipidemia and arterial hypertension were taken into account.

Treatment with DAAs provide several benefits for CKD patients with HCV infection: reduction in cardiovascular morbidity and mortality, moreover progression of renal dysfunction can be achieved, reducing liver disease progression, and improving patients’ well-being.

It was demonstrated that DAA therapy can improve renal function in HCV patients, particularly those with baseline eGFR below 60 ml/min per1.73m². This finding suggests that early intervention with DAAs may help prevent further deterioration of kidney function, emphasizing the need for timely diagnosis and treatment of HCV infection in patients with chronic kidney disease.

Data is discussed well and is of interest to a wide range of readers

I would recommend minor revisions.

  1. Direct-acting antivirals used for the treatment are not mentioned. Did the authors mean to analyze the effect regardless of the specific antiviral drug?

References 46-48 seem strange

Thank you very much for this comment. All patients were treated with direct-acting antiviral drugs following the guidelines recommended by the APASL. Treatment considerations included factors such as HCV genotype, viral loads, and viral response, with a standard treatment duration of 12–24 weeks as stipulated by the reimbursement guidelines of the Taiwan National Health Insurance. Also we have added this information in Materials and Methods section.

We have revised the References in our manuscript and now it has 39 references.  

Round 2

Reviewer 1 Report

Comments and Suggestions for Authors

The similarity index is unacceptably high and must be reduced to meet acceptable standards. Before submitting, a plagiarism detection program must be applied.

Author Response

1. The similarity index is unacceptably high and must be reduced to meet acceptable standards. Before submitting, a plagiarism detection program must be applied.

Thank you for this comment. For similarity, we re-edited and developed our manuscript according to the reviewers' recommendations.
